# Relationship between Drug Holidays of Antiresorptive Agents and Surgical Outcomes in Cancer Patients with Medication-Related Osteonecrosis of the Jaw

**DOI:** 10.3390/ijerph19084624

**Published:** 2022-04-12

**Authors:** Mitsunobu Otsuru, Sakiko Soutome, Keisuke Omori, Koki Suyama, Kota Morishita, Saki Hayashida, Maho Murata, Yukinori Takagi, Miho Sasaki, Misa Sumi, Yuka Kojima, Shunsuke Sawada, Yuki Sakamoto, Masahiro Umeda

**Affiliations:** 1Department of Clinical Oral Oncology, Nagasaki University Graduate School of Biomedical Sciences, Nagasaki 852-8588, Japan; bb55315203@ms.nagasaki-u.ac.jp (K.O.); koki.s307@nagasaki-u.ac.jp (K.S.); k-morishita@nagasaki-u.ac.jp (K.M.); sakihaya@nagasaki-u.ac.jp (S.H.); mamurata@nagasaki-u.ac.jp (M.M.); mumeda@nagasaki-u.ac.jp (M.U.); 2Department of Oral Health, Nagasaki University Graduate School of Biomedical Sciences, Nagasaki 852-8588, Japan; sakiko@nagasaki-u.ac.jp; 3Department of Radiology and Biomedical Informatics, Nagasaki University Graduate School of Biomedical Sciences, Nagasaki 852-8588, Japan; yuki@nagasaki-u.ac.jp (Y.T.); sasaki-m@nagasaki-u.ac.jp (M.S.); misa@nagasaki-u.ac.jp (M.S.); 4Department of Dentistry and Oral Surgery, Kansai Medical University, Osaka 852-8588, Japan; yukojimayk@gmail.com (Y.K.); sawadash@hirakata.kmu.ac.jp (S.S.); s.yukioutdoor@gmail.com (Y.S.)

**Keywords:** medication-related osteonecrosis of the jaw, cancer, drug holiday, outcome

## Abstract

It is controversial as to whether the withdrawal of antiresorptive (AR) agents is necessary while treating medication-related osteonecrosis of the jaw (MRONJ). In this study, we investigated whether a drug holiday promoted sequestrum separation and improved the surgical outcomes of MRONJ patients with malignant tumors, who were undergoing high-dose AR therapy. In total, we included 103 MRONJ patients with malignant tumors as their primary disease who underwent surgery at Nagasaki University Hospital or Kansai Medical University Hospital from January 2009 to December 2020. We recorded the patients’ age, sex, primary disease, MRONJ stage, type and administration period of the AR agent, presence of diabetes, corticosteroid use, drug holiday period, white blood cell count, serum albumin, serum creatinine, outcomes, and computed tomography findings. The relationships between a drug holiday and sequestrum separation, and between a drug holiday and outcome, were analyzed. Drug holidays of 60, 90, and 120 days were not significant factors of sequestrum separation and did not influence patients’ surgical outcomes as per the univariate and multivariate analyses. MRONJ patients with cancer as their primary disease should be operated upon immediately and without drug holidays if their general condition permits surgery.

## 1. Introduction

Medication-related osteonecrosis of the jaw (MRONJ) is known to be one of the severe side effects of antiresorptive (AR) agents, such as bisphosphonate (BP) and denosumab (Dmab) [1,2]. The position papers of the American Association of Oral and Maxillofacial Surgeons (AAOMS) [2] and the Japanese Society of Oral and Maxillofacial Surgeons (JSOMS) [3] recommended a conservative therapy, such as a combination of antimicrobial mouthwash and systemic antibiotics for stage II MRONJ. For refractory cases, a combination of multiple antimicrobial agents, long-term antimicrobial administration, and the continuous administration of intravenous antimicrobial agents was recommended. The removal of the sequestrum, curettage of necrotic bones, and osteotomy were also recommended for intractable cases. 

Some systematic reviews have shown that the healing rate of MRONJ patients who undergo surgery is higher than that of those who receive conservative therapy [4,5,6]. Hayashida et al. [7], in a multicenter retrospective study of 361 patients with MRONJ, reported that the complete healing rate of 159 patients who underwent surgical treatment was 76.7%, but that of 202 patients who received non-surgical treatment was only 25.2%. Furthermore, when background factors of the surgical and non-surgical cases were adjusted by the propensity score matching method, the cure rates of the surgical and non-surgical cases were 68.2% and 25.0%, respectively.

Thus, it is unclear whether a drug holiday of an AR agent is necessary before surgical treatment. The position papers of both AAOMS and JSOMS suggested that AR therapy may need to be discontinued until the MRONJ treatment is complete, although a drug holiday is not recommended for patients with high fracture risk or progressive bone metastases [2,3]. However, there is no evidence regarding the effectiveness or ideal duration of a drug holiday of AR agents. Therefore, we investigated whether a drug holiday of AR agents contributed to the formation of sequestrum separation and treatment outcomes in MRONJ patients whose primary disease was malignancy.

## 2. Materials and Methods

### 2.1. Study Design and Patients

This was a retrospective observational study. From January 2009 to December 2020, 70 out of 175 MRONJ patients with malignant tumors, who underwent treatment at Nagasaki University Hospital or Kansai Medical University Hospital, received only conservative therapy. Of the 105 patients who underwent surgery, 103 were included in the study, excluding two patients whose data could not be collected.

### 2.2. Variables

We recorded the following patient data: age, sex, primary disease, MRONJ site, MRONJ stage, type of AR agent, administration period of AR agent, presence of diabetes, corticosteroid use, drug holiday period, white blood cell count, serum albumin, serum creatinine, surgical procedure, outcomes, and computed tomography (CT) findings. The primary disease category was divided into solid cancer and multiple myeloma. The MRONJ stage was determined at the time of initial diagnosis and according to the AAOMS (2014) [2]. Regarding the drug holiday, both time periods from the final scheduled administration date after the last dose to CT imaging and the period until surgery were examined. Our MRONJ surgery is based on the complete resection of the necrotic bone and one layer of surrounding healthy bone, followed by primary suture. Only the resection of necrotic bone was not performed (so-called conservative surgery). The surgical procedure was divided into marginal resection (partial maxillectomy or marginal mandibulectomy) and segmental resection (segmental mandibulectomy). CT findings were evaluated through a consensus between the oral surgeon and radiologist, using preoperative images for the presence or absence of the extent of the osteolytic area (localized or extended) and periosteal reaction. Osteolysis was classified as localized (the maxilla does not extend to the floor of the nasal cavity or the maxillary sinus floor, and the mandible extends above the mandibular canal) or extended (the maxilla extends to the floor of the nasal cavity or maxillary sinus floor, and the mandible includes the lower mandibular canal). Sequestrum separation was defined as a clear and almost total separation between the sequestrum and the normal bone (Figure 1).

Postoperative follow-up was based on monthly medical examinations and CT images every three months. Patients were asked to visit the hospital as per their general condition. The treatment outcome was divided into healing and non-healing. Healing was defined as a lack of osteolysis progression on the CT images, and the disappearance of all clinical symptoms, including bone exposure, pain, and infection, and the date of healing, was registered. We examined the relationship between a preoperative drug holiday of AR agents and treatment outcome in terms of healing time, as well as the relationship between the drug holidays and sequestrum separation.

### 2.3. Statistical Analyses

All statistical analyses were performed using SPSS software (version 26.0; Japan IBM Co., Ltd., Tokyo, Japan), and a two-tailed *p*-value of ˂ 0.05 was considered significant. The correlation between each variable and sequestrum separation was analyzed using the one-way ANOVA for continuous variables and Fisher’s exact test for categorical variables. The correlation between each variable and treatment outcome was analyzed using univariate and multivariate cox regression analyses. The relationship between a drug holiday and the treatment outcome was illustrated using the Kaplan–Meier method.

### 2.4. Ethical Approval

The study protocol conformed to the ethical guidelines of the Declaration of Helsinki and the Ethical Guidelines for Medical and Biological Research Involving Human Subjects by the Ministry of Health, Labor, and Welfare of Japan. This study was approved by the Institutional Review Board (#21021509) of Nagasaki University Hospital. The need for informed consent was waived by the IRB of Nagasaki University Hospital. This was a retrospective study; thus, the research plan was published on the homepage of the participating hospitals according to the instructions of the IRB, and in accordance with the guaranteed opt-out opportunity.

## 3. Results

### 3.1. Patient Characteristics

The number of MRONJ patients whose primary disease was a malignancy and who underwent surgical treatment was 103 (Table 1). Nine patients (8.7%) showed sequestrum separation and ninety-four (91.3%) did not. We encouraged oncologists to not withdraw AR agents for the purpose of treating MRONJ, but to administer them (if necessary) for the treatment of the primary disease. However, several patients had drug holidays of AR agents at the discretion of their oncologists. There were 22 patients who had drug holidays for more than 60 days before the CT examination, 17 patients for more than 90 days, and 11 patients for more than 120 days. Owing to the waiting period between the first visit and surgery, there were 27, 21, and 15 patients of drug holidays for 60, 90, and 120 days, respectively, at the time of surgery. Postoperatively, patients who were administered AR agents continued to take them, while those who took drug holidays were not administered these agents until healing was observed.

### 3.2. Factors Related to Separation of Sequestrum

Sequestrum separation was found in 9 out of 103 patients (8.7%). There were no factors related to the separation. A drug holiday before CT for more than 60 days (*p* = 0.095), 90 days (*p* = 0.641), or 120 days (*p* = 0.246) did not become a significant factor of sequester separation (Table 2).

### 3.3. Relationship between a Drug Holiday before Surgery and Treatment Outcome

A univariate analysis revealed no significant factors related to the treatment outcome. According to a multivariate analysis using stepwise selection, the use of steroid (*p* = 0.043), lower serum creatinine (*p* = 0.003), and localized osteolytic area (*p* = 0.014) were significantly correlated with poor treatment outcome (Table 3). Figure 2 shows the relationship between a drug holiday before surgery and the treatment outcome using the Kaplan–Meier method. A drug holiday before surgery of 60 days (*p* = 0.220), 90 days (*p* = 0.394), or 120 days (*p* = 0.178) did not improve treatment outcomes.

## 4. Discussion

This study showed that drug holidays of 60, 90 and 120 days were not significant factors of sequestrum separation, and did not influence surgical outcomes in cancer patients with MRONJ undergoing surgical treatment.

It is unclear whether surgical treatment or conservative treatment is the better strategy for MRONJ. In a retrospective multicenter study, the authors reported the superiority of surgical therapy over conservative therapy [7]. Therefore, at our institution, surgery is the first-line treatment for MRONJ, regardless of whether the underlying disease is osteoporosis or cancer. There is indeed a difference in the backgrounds of patients with malignancy and osteoporosis as the primary disease. Patients with malignant tumors use higher doses of AR agents and are also more likely to have weakened immunity due to anticancer drugs and poor general health. Cancer can progress rapidly; thus, we immediately try to perform surgery for MRONJ without any drug holiday after consulting patients and oncologists.

The AAOMS position paper recommended a 3-month drug holiday of a low-dose AR agent before invasive dental surgery in 2009 [8], but the American Dental Association Council on Scientific Affairs revised their prior recommendation of a drug holiday and suggested that patients receiving lower cumulative doses of BP (<2 years) or denosumab could continue AR therapy during invasive dental treatment in 2011 [9]. Some investigators suggested that a drug holiday should be considered in patients who received an oral AR agent for more than 4 years, and others advocated a short drug holiday [6,10]. Currently, the revised position papers of AAOMS [2], JSOMS [3] and the Korean Association of Oral and Maxillofacial Surgeons (KAOMS) [11] recommend a 2-month drug holiday for a low-dose AR agent before tooth extraction in patients with long BP exposure (>4 years) or those with comorbid risk factors, such as rheumatoid arthritis, glucocorticoid exposure, diabetes, and smoking.

It is often not appropriate to withdraw the AR agent therapeutically in cancer patients; therefore, the position papers of AAOMS and JSOMS suggested that invasive dental treatments should be avoided [2,3]. However, these position papers also recommended that, if MRONJ develops, the oncologist may consider discontinuing AR therapy until soft tissue closure occurs, depending on the disease status. Hayashida et al. [12] reported that preoperative drug holidays do not improve surgical outcomes in MRONJ patients treated with low or high doses of an AR agent. Although some oral surgeons may consider surgery to be easy and minimally invasive if a drug holiday promotes sequestrum separation, there have been no studies clarifying the relationship between a drug holiday and sequestrum separation.

BP is bound tightly to the bone matrix, and osteoclasts take up BP into the cytoplasm during bone resorption, subsequently leading to apoptosis. BP may remain in the bone for months or years, and the bone resorption inhibitory effect lasts for a long time. Dmab is a monoclonal antibody which selectively binds to RANKL. Dmab is not absorbed in the bone tissue, but circulates in the blood. Dmab prevents the formation, function, and survival of osteoclasts, and has a half-life of 26 days in the blood [13,14]. From the results of these basic studies, it can be considered that the withdrawal of AR agents is more effective with Dmab. However, Hasegawa reported that in patients with osteoporosis and malignant tumors, drug holidays of 2 to 3 months did not reduce the risk of MRONJ development after tooth extraction [15,16,17].

It is not clear how long AR agents should be withdrawn to recover from osteoclast inhibition. Recently, we examined surgical materials of osteoporosis patients receiving oral BP immunohistologically and found that osteoclast morphology did not recover with a drug holiday of less than 12 months [18]. Although the patients receiving high-dose AR agents have not been sufficiently investigated, it is considered that the osteoclast function may not be recovered after several months of drug suspension. In this study, the authors examined the benefits of preoperative drug holidays, specifically in MRONJ patients with malignancy as the primary disease. The sequestrum separation was 8.7% even after 3 months of drug holiday, and drug holidays of AR agents had little effect on sequestrum separation. Additionally, surgery with a drug holiday has no effect on the improvement of outcomes for MRONJ patients. Although we do not believe that the withdrawal of bone resorption inhibitors is necessary before and during MRONJ surgery, some patients were withdrawn by their oncologists prior to their first visit to our department. Withdrawal is difficult in patients with rapidly progressing bone metastases; thus, we considered that the withdrawal cases included many patients whose cancer had progressed relatively slowly and whose general condition was good. Nevertheless, there was no difference in treatment outcomes between patients who withdrew and those who continued the treatment, suggesting that drug withdrawal does not contribute to improved treatment outcomes. Considering these factors, MRONJ patients with cancer as the primary disease should be operated upon immediately without a drug holiday if their general condition is relatively good, clinical symptoms are strong, and quality of life is improved.

There are some limitations in the study. Firstly, this is a retrospective analysis using a small number of patients; therefore, it is unclear whether the results obtained can be generalized. Secondly, this study was a cross-sectional study and did not longitudinally observe the effects of a drug holiday. However, to the best of our knowledge, this is the first study to investigate how a preoperative drug holiday affects sequestrum separation and treatment outcomes in patients receiving high-dose AR agent therapy. In future, studies with larger sample sizes and more detailed examinations should be conducted to confirm our results.

## 5. Conclusions

Drug holidays of 60, 90 and 120 days were not significant factors for sequestrum separation and did not influence the surgical outcomes of cancer patients with MRONJ undergoing surgical treatment. MRONJ patients with cancer as their primary disease should be operated upon immediately without drug holidays if their general condition permits surgery.

## Figures and Tables

**Figure 1 ijerph-19-04624-f001:**
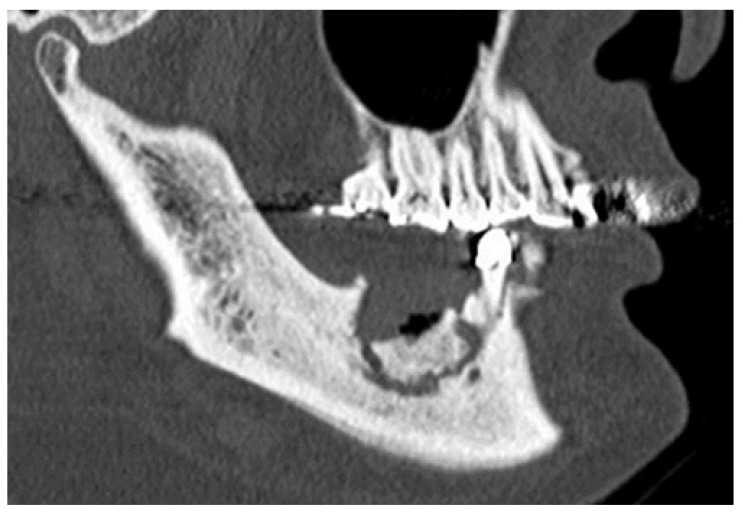
A computed tomography 3D image. The sequestrum separation was defined as a clear and almost total separation between the sequestrum and the normal bone.

**Figure 2 ijerph-19-04624-f002:**
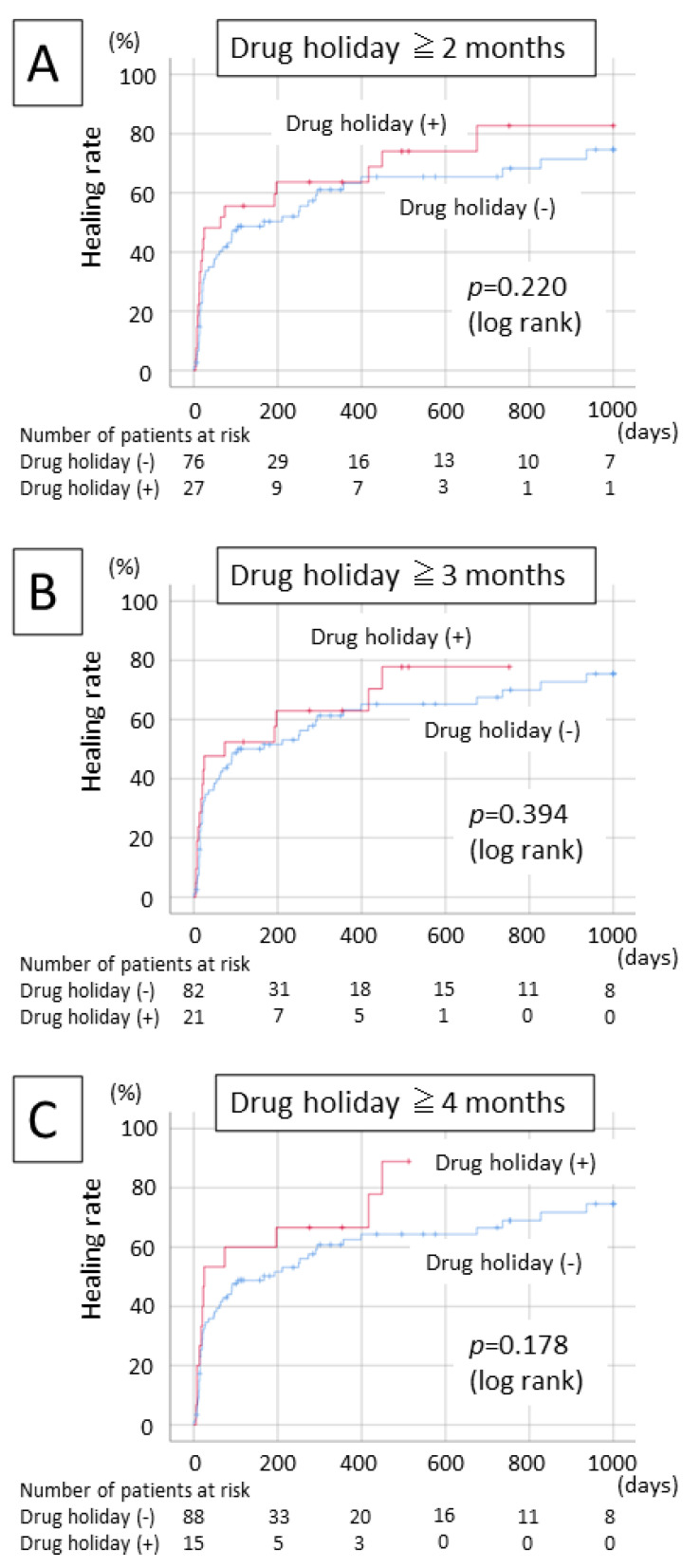
Healing rate by drug holiday; (**A**) drug holiday for 60 days; (**B**) drug holiday for 90 days; and (**C**) drug holiday for 120 days.

**Table 1 ijerph-19-04624-t001:** Patient characteristics.

Variable		Number of Patients/Mean ± SD
Sex	man	45
	woman	58
Age	(years)	70.3 ± 10.7
Primary disease	solid cancer	88
	myeloma	15
MRONJ site	upper jaw	39
	lower jaw	64
MRONJ stage	stage 1	11
	stage 2	70
	stage 3	22
Antiresorptive agent	BP	37
	Dmab	42
	BP→Dmab	24
Use of steroid	(−)	81
	(+)	22
Diabetes	(−)	79
	(+)	24
Duration of administration	<4 years	67
	≧4 years	34
	unknown	2
Duration of drug holiday before CT examination	(days)	34.1 ± 78.0
	≧60 days	22
	≧90 days	17
	≧120 days	11
Duration of drug holiday before surgery	(days)	69.2 ± 171.9
	≧60 days	27
	≧90 days	21
	≧120 days	15
Leukocytes	(/µL)	6387 ± 1813
Albumin	(g/dL)	3.69 ± 0.522
Creatinine	(mg/dL)	0.889 ± 0.396
Extent of osteolytic area	localized	75
	extended	28
Periosteal reaction	(−)	75
	(+)	28
Separation of sequestrum	(−)	94
	(+)	9
Surgical procedure	marginal resection	94
	segmental resection	9
Total		103

Abbreviation: MRONJ: medication-related osteonecrosis of the jaw; BP: bisphosphonate; Dmab: denosumab; CT: computed tomography.

**Table 2 ijerph-19-04624-t002:** Relationship between each variable and sequestrum separation (univariate analysis).

Variable		Separation of Sequestrum (−)	Separation of Sequestrum (+)	*p*-Value
Sex	man	42	3	0.728
	woman	52	6	
Age		70.3 ± 10.7	70.2 ± 9.30	0.993
Primary disease	solid cancer	81	7	0.616
	myeloma	13	2	
MRONJ site	upper jaw	35	4	0.727
	lower jaw	59	5	
MRONJ stage	stage 1	9	2	0.424
	stage 2	64	6	
	stage 3	21	1	
Antiresorptive agent	BP	33	4	0.482
	Dmab	40	2	
	BP→Dmab	21	3	
use of steroid	(−)	72	9	0.199
	(+)	22	0	
Diabetes	(−)	72	7	1.000
	(+)	22	2	
Duration of administration	<4 years	64	5	0.473
	≧4 years	30	4	
Drug holiday before CT ≧ 60 days	(−)	76	5	0.095
	(+)	18	4	
Drug holiday before CT ≧ 90 days	(−)	79	7	0.641
	(+)	15	2	
Drug holiday before CT ≧ 120 days	(−)	85	7	0.246
	(+)	9	2	
Leukocytes	(/µL)	6192 ± 1854	6120 ± 1150	0.932
Albumin	(g/dL)	3.67 ± 0.529	3.84 ± 0.441	0.392
Creatinine	(mg/dL)	0.912 ± 0.396	0.908 ± 0.413	0.975
Extent of osteolytic area	localized	70	5	0.250
	extended	24	4	
Periosteal reaction	(−)	70	5	0.250
	(+)	24	4	
Surgical procedure	marginal resection	90	8	0.373
	segmental resection	4	1	

Abbreviation: MRONJ: medication-related osteonecrosis of the jaw; BP: bisphosphonate; Dmab: denosumab; CT: computed tomography.

**Table 3 ijerph-19-04624-t003:** Relationship between each variable and treatment outcome (univariate and multivariate cox regression).

Variable		Univariate Analysis			Multivariate Analysis		
		*p*-Value	HR	95% CI	*p*-Value	HR	95% CI
Sex	woman/man	0.765	0.93	0.579–1.492			
Age	(years)	0.062	0.997	0.975–1.019			
Primary disease	multiple myeloma/solid cancer	0.769	0.911	0.486–1.704			
MRONJ site	lower jaw/upper jaw	0.083	0.653	0.403–1.058			
MRONJ stage	stage 3/2/1	0.59	0.892	0.590–1.350			
Antiresorptive agent	BP→Dmab/Dmab/BP	0.22	0.821	0.600–1.125			
use of steroid	(+)/(−)	0.408	0.788	0.449–1.385	0.043	0.494	0.250–0.977
Diabetes	(+)/(−)	0.208	1.398	0.830–2.354			
Duration of administration	≧4 years/<4 years	0.918	0.974	0.590–1.608			
Drug holiday before surgery ≧90 days	(+)/(−)	0.398	1.281	0.720–2.283			
Leukocytes	(/µL)	0.124	1	0.999–1.000			
Albumin	(g/dL)	0.068	0.659	0.421–1.032			
Creatinine	(mg/dL)	0.074	1.789	0.946–3.383	0.003	2.787	1.410–5.509
Extent of osteolytic area	Extended/localized	0.22	1.387	0.822–2.341	0.014	2.233	1.176–4.238
Periosteal reaction	(+)/(−)	0.266	0.727	0.415–1.274			
Separation of sequestrum	(+)/(−)	0.532	1.308	0.564–3.032			
Surgical procedure	segmental resection/margical resection	0.605	1.307	0.474–3.601			

Abbreviation: MRONJ: medication-related osteonecrosis of the jaw; BP: bisphosphonate; Dmab: denosumab.

## Data Availability

The datasets used and analyzed during the study are available from the corresponding author upon reasonable request.

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
