# Peer review of "Relationship between Drug Holidays of Antiresorptive Agents and Surgical Outcomes in Cancer Patients with Medication-Related Osteonecrosis of the Jaw"

_ijerph, 2022, doi:10.3390/ijerph19084624_

Round 1
Reviewer 1 Report
The article must be submitted to a professional copy editing service as many paragraphs are not very well written (Eg. “ the maxilla extends to the floor of the nasal cavity” ) and there are a few grammatical errors.
Otherwise, I believe the article has some flaws; I have attached my other comments below.
- The authors state that this is a retrospective evaluation that compared patients that underwent a drug holiday to those that did not. I believe the authors should explain on what basis this decision was made. They also stated that patients that had a worse clinical condition were immediately treated and no drug holiday was respected. IF that is the case, the findings from this study are severely flawed as patients that followed a drug holiday were expected to have a worse clinical outcome, simply on the basis of their previous condition. Presenting the data of the two groups separated might help the reader understand on its own the weight this occurrence might have had.
- The disparity between the two samples is another major flaw, as only 27 patients followed a drug holiday between the surgical therapy and only 22 before the CT.
Author Response
To Reviewer #1
- The article must be submitted to a professional copy editing service as many paragraphs are not very well written (Eg. “ the maxilla extends to the floor of the nasal cavity” ) and there are a few grammatical errors.
(Reply)
Revised manuscript was submitted to a professional English editing company (Editage).
- The authors state that this is a retrospective evaluation that compared patients that underwent a drug holiday to those that did not. I believe the authors should explain on what basis this decision was made. They also stated that patients that had a worse clinical condition were immediately treated and no drug holiday was respected. IF that is the case, the findings from this study are severely flawed as patients that followed a drug holiday were expected to have a worse clinical outcome, simply on the basis of their previous condition. Presenting the data of the two groups separated might help the reader understand on its own the weight this occurrence might have had.
(Reply)
The next description was added in ‘Discussion’.
Although we do not believe that withdrawal of bone resorption inhibitors is necessary before and during MRONJ surgery, some patients had been withdrawn by their oncologists prior to their first visit to our department. Since withdrawal is difficult in patients with rapidly progressing bone metastases, we considered that the withdrawal cases included many patients whose cancer had progressed relatively slowly and whose general condition was good. Nevertheless, there was no difference in treatment outcome between patients who withdrew and those who continued treatment, suggesting that drug withdrawal does not contribute to improved treatment outcome.
- The disparity between the two samples is another major flaw, as only 27 patients followed a drug holiday between the surgical therapy and only 22 before the CT.
(Reply)
I'm sorry for the incomprehensible expression. Twenty-two patients had a drug holiday of 60 days or more before CT examination. Including the period from CT examination to surgery, 27 patients had a drug holiday of 60 days or more before surgery.
The items listed in Table 1 have been revised.

Reviewer 2 Report
Kudos to the authors for their hard work.
Because I am also one of the clinicians who are interested in MRONJ, I made a careful review.
1. Why did the authors use the term 'sequester'? Is it an official term for the mass of sequestrum? I could not find this term in other articles related to MRONJ. Why did not the authors use the term 'sequestration' or 'separation of sequetrum'? I think it is better for authors to replace.
2. There is no comment about surgical procedure that authors performed. I wonder whether the surgical procedures consisted of just sequestrectomy or whether saucerization or radical excision was performed. And also, I wonder if primary closure was obtained in all surgeries performed by the authors. Because it may affect the treatment outcomes (healing or non-healing) defined by the authors.
3. As the preoperative drug holiday is an important factor, the administration of antiresorptive agents during healing period after surgery is also a very important factor. However, the authors make no mention of if anywhere in the manuscript.
Author Response
To Reviewer #2
- Why did the authors use the term 'sequester'? Is it an official term for the mass of sequestrum? I could not find this term in other articles related to MRONJ. Why did not the authors use the term 'sequestration' or 'separation of sequetrum'? I think it is better for authors to replace.
(Reply)
The term ‘sequester’ in the manuscript and Tables was corrected to ‘sequestrum’.
- There is no comment about surgical procedure that authors performed. I wonder whether the surgical procedures consisted of just sequestrectomy or whether saucerization or radical excision was performed. And also, I wonder if primary closure was obtained in all surgeries performed by the authors. Because it may affect the treatment outcomes (healing or non-healing) defined by the authors.
(Reply)
Our MRONJ surgery is based on complete resection of necrotic bone and one layer of surrounding healthy bone, followed by primary suture. We have not performed only resection of necrotic bone (so-called conservative surgery). Surgical procedure was divided into marginal resection (partial maxillectomy or marginal mandibulectomy) and segmental resection (segmental mandibulectomy). The descriptions were added in ‘Materials and methods’. Further, they were added to the statistical analysis (Table 1-3).
- As the preoperative drug holiday is an important factor, the administration of antiresorptive agents during healing period after surgery is also a very important factor. However, the authors make no mention of if anywhere in the manuscript.
Postoperatively, patients who had been on AR agents were continued on them, while those who had been off them remained off them until healing was observed. The descriptions were added in ‘Results’.

Reviewer 3 Report
I have to congratulate the authors of this work. Although the limited sample stands out in the limitations, I think they have enough power to extrapolate the results.
It is very useful for clinicians and has been very easy to understand and read.
I would like the authors to clarify if any patient continued with the treatment, without any type of drug holiday. It is an aspect that has remained unclear to me. However, the way of presenting the results in the range of 60, 90 and 120 days seems correct to me, taking into account the literature and the clinical practice guidelines, and I believe that it is well defended in the discussion.
Simply for the authors to point out that no patient continued medication during surgery (if that is the case).
Author Response
To Reviewer #3
- I would like the authors to clarify if any patient continued with the treatment, without any type of drug holiday. It is an aspect that has remained unclear to me. However, the way of presenting the results in the range of 60, 90 and 120 days seems correct to me, taking into account the literature and the clinical practice guidelines, and I believe that it is well defended in the discussion. Simply for the authors to point out that no patient continued medication during surgery (if that is the case).
(Reply)
Basically, we encourage oncologists not to withdraw AR agents for the purpose of treating MRONJ, but to administer them if necessary for the treatment of the primary disease. The descriptions were added in ‘Results’.

Round 2
Reviewer 1 Report
The authors responded properly to my comments.
Reviewer 2 Report
I thank the authors for their sincere responses. I would like to ask for your continuous research on ONJ.